# Targeting SARS-CoV-2 nsp13 Helicase and Assessment of Druggability Pockets: Identification of Two Potent Inhibitors by a Multi-Site In Silico Drug Repurposing Approach

**DOI:** 10.3390/molecules27217522

**Published:** 2022-11-03

**Authors:** Isabella Romeo, Francesca Alessandra Ambrosio, Giosuè Costa, Angela Corona, Mohammad Alkhatib, Romina Salpini, Saverio Lemme, Davide Vergni, Valentina Svicher, Maria Mercedes Santoro, Enzo Tramontano, Francesca Ceccherini-Silberstein, Anna Artese, Stefano Alcaro

**Affiliations:** 1Dipartimento di Scienze della Salute, Università degli Studi “Magna Græcia” di Catanzaro, Campus “S. Venuta”, Viale Europa, 88100 Catanzaro, Italy; 2Net4Science Academic Spin-Off, Università degli Studi “Magna Græcia” di Catanzaro, Campus “S. Venuta”, Viale Europa, 88100 Catanzaro, Italy; 3Dipartimento di Medicina Sperimentale e Clinica, Università degli Studi “Magna Græcia” di Catanzaro, Campus “S. Venuta”, Viale Europa, 88100 Catanzaro, Italy; 4Department of Life and Environmental Sciences, University of Cagliari, Cittadella Universitaria di Monserrato, 09124 Cagliari, Italy; 5Dipartimento di Medicina Sperimentale, Università Tor Vergata di Roma, Via Montpellier, 1, 00133 Roma, Italy; 6Istituto per le Applicazioni del Calcolo “Mauro Picone”-CNR, 00185 Rome, Italy

**Keywords:** SARS-CoV-2 (COVID-19), helicase, drug repurposing, nsp13, conservation analysis, inhibitory activity

## Abstract

The SARS-CoV-2 non-structural protein 13 (nsp13) helicase is an essential enzyme for viral replication and has been identified as an attractive target for the development of new antiviral drugs. In detail, the helicase catalyzes the unwinding of double-stranded DNA or RNA in a 5′ to 3′ direction and acts in concert with the replication–transcription complex (nsp7/nsp8/nsp12). In this work, bioinformatics and computational tools allowed us to perform a detailed conservation analysis of the SARS-CoV-2 helicase genome and to further predict the druggable enzyme’s binding pockets. Thus, a structure-based virtual screening was used to identify valuable compounds that are capable of recognizing multiple nsp13 pockets. Starting from a database of around 4000 drugs already approved by the Food and Drug Administration (FDA), we chose 14 shared compounds capable of recognizing three out of four sites. Finally, by means of visual inspection analysis and based on their commercial availability, five promising compounds were submitted to in vitro assays. Among them, PF-03715455 was able to block both the unwinding and NTPase activities of nsp13 in a micromolar range.

## 1. Introduction

As of 16 September 2022, the WHO had already received reports of more than 608 million confirmed cases of COVID-19, with more than 6 million deaths worldwide [1]. Unfortunately, the number of infections continues to rise, as does the number of re-infections, mainly due to the potentiality of SARS-CoV-2 to produce mutant strains that are able to evade neutralizing antibodies [2,3]. Many of the vaccines used to target the spike protein [4], but the presence of several mutations have prevented protection against all newly emerging strains [5]. In addition, millions of immune-compromised patients may not get full protection after vaccination. On the other hand, therapeutic options for COVID-19 are also currently limited. In the past, other human coronaviruses (CoVs) have been treated with antiviral drugs that, have been proved to be ineffective due to structural differences in SARS-CoV-2 as compared to other human CoVs. Moreover, the mechanism of the pathogenesis of this virus is very complex [6]; thus, the research efforts have been devoted to providing deep insights into the SARS-CoV-2 genome in order to delineate the viral mechanism of action and to identify novel drugs that are capable of blocking viral entry or propagation. Nevertheless, the waiting time required for de novo development to be realized is not feasible due to the emergency situation the world has been dealing with in the last two years. Therefore, the repurposing of approved or investigational drugs could represent a practical approach for the fast identification, characterization, and deployment of antiviral treatments, thanks to the availability of existing detailed data on drug medicinal chemistry, human pharmacology, and toxicology [7,8]. In this direction, several clinical trials have suggested different therapeutic options for COVID-19 treatment, including various drugs that have been approved for other indications such as antimalarial drugs (hydroxychloroquine and chloroquine), antiparasitic drugs (ivermectin), anti-inflammatory corticosteroids (dexamethasone and prednisolone), antibacterial (azithromycin), antivirals (lopinavir, ritonavir, remdesivir, molnupiravir), antihypertensives (losartan), and immunomodulators [9,10]. Among them, remdesivir, favipiravir, and molnupiravir, which have already been used as broad-spectrum experimental antiviral agents [11,12,13], have been highlighted as SARS-CoV-2 RNA-dependent RNA polymerase (RdRp) inhibitors and have been associated with a remarkable improvement in disease outcomes and a decrease in recovery time. However, some studies have demonstrated that both remdesivir and molnupiravir could increase the frequency of viral RNA mutations, which cause the emergence of new SARS-CoV-2 strains associated with resistance [14,15,16,17].

By contrast, among the 16 non-structural proteins (nsp1–16) that collectively form the machinery for viral replication and transcription, the helicase enzyme (nsp13) is characterized by a high degree of genetic conservation across the different domains and among the different SARS-CoV-2 variants (>99.0%) [18,19]. Nsp13 belongs to the helicase super-family 1B and catalyzes the unwinding of double-stranded DNA or RNA in a 5′ to 3′ direction by exploiting the energy of nucleotide triphosphate hydrolysis. Along the two “RecA like” subdomains 1A and 2A, which are responsible for nucleotide binding and hydrolysis, nsp13 contains three additional domains: the N-terminal zinc binding domain, involved in the coordination of three structural zinc ions, the helical “stalk” domain, and the beta-barrel 1B domain.

Previous studies have demonstrated that the nsp13 helicase forms a stable complex with the replication–transcription complex (RTC) (nsp7/nsp8/nsp12). More specifically, it is involved in a crucial interaction with the viral RNA-dependent RNA polymerase nsp12 [20,21], thus stimulating helicase activity by means of mechano-regulation. Moreover, this enzyme is also involved in the formation of the viral 5′ mRNA cap due to its RNA 5′ triphosphatase activity within the same active site [22]. In this work, we performed a detailed conservation analysis of the SARS-CoV-2 helicase, and we applied computational and bioinformatic tools in order to predict the enzyme-binding pockets that were to be investigated by an in silico drug repurposing approach. Among the 14 selected investigational and approved compounds by the Food and Drug Administration (FDA), 5 were purchased, and 2 were found to effectively inhibit the enzyme in a micromolar range.

## 2. Results

### 2.1. Conservation Analysis

The conservation analysis along with a comprehensive examination of SARS-CoV-2 helicase residues are crucial prior to investigating drug compounds and identifying druggable targets. The analysis was performed by using over 5.1 million high-quality SARS-CoV-2 sequences. The overall analysis revealed a high degree of conservation in the helicase. In particular, 54.2% (326/601) of residues never mutated, and 44.4% (267/601) were extremely conserved in >99.0% of the sequences analyzed. Among the remaining eight residues, seven showed a substantial degree of conservation (ranging from 95.5% to 98.6% of the sequences analyzed). Only one residue at position 77 was found to have mutated in 60% of the sequences, mainly as a consequence of the mutation Pro77Leu, which represents the wild-type amino acid in the previously circulating variant of concern Delta and all its sublineages (Figure 1).

More importantly, out of 601 residues characterizing the helicase, 101 residues were of great interest due to their localization in crucial domains and their ability to efficiently bind the tested compounds. The analysis revealed that 70.3% (71/101) of residues never mutated, while 24.8% (25/101) were extremely conserved, with no mutations in >99.9% of the sequences analyzed. The remaining five residues showed a degree of conservation ranging from 98.6% to 99.9% (Appendix A), which again supports the high degree of conservation of the helicase, particularly in domains crucial for enzymatic activity. This reinforces the role of nsp13 as an appealing target for antiviral drugs.

### 2.2. Identification of the Druggable Binding Pockets

As illustrated in Figure 2A, the GRID software [23] allowed us to identify three cavities: pocket 1 (in pink), pocket 2 (in violet), and pocket 3 (in orange).

According to the literature, pocket 1 and pocket 2 are the most conserved sites of the enzyme and allow for the accommodation of the ATP nucleotide and the RNA strand, respectively [24]. In fact, pocket 1 plays a crucial role in the translocation of genetic material since the interaction, hydrolysis, and release of ATP triggers conformational changes in the enzyme and remodels the interface of the RNA. Pocket 2 refers to the binding channel of the RNA/DNA strand. Another cavity, pocket 3, was identified between the *n*-terminal portion involved in zinc binding (ZBD) and the stalk domain, whose contribution to the enzymatic activity of SARS-CoV-2 nsp13 is relevant. This is supported by the evidence that the deletion of the stalk domain as well as the linker insertion between the stalk and helicase domains abrogate the ATPase activity of the enzyme and reduce its stability, respectively [25]. In addition to the predicted pockets, we also included the site at the interface between the ZBD domain and the non-structural protein 8 (nsp8), which we refer to as pocket 4 (Figure 2B). This connection forms a stable complex that is responsible for generating backtracked RTC for proofreading and template switching during sub-genomic RNA transcription [20].

### 2.3. Residue Interaction Network Analysis of Helicase

Although the search for the binding sites of a protein relies on contact energy minimization methods [23], we identified many residues that belong to active pockets by means of a conformational investigation based on the helicase Residue Interaction Network (RIN) [26] and classical network centrality measures such as Residue Centrality Analysis (RCA), Betweenness Centrality Analysis (BCA), and Closeness Centrality Analysis (CCA).

The statistical relevance of the result was demonstrated by the very low *p*-value associated with the enrichment analysis of the common residues, i.e., the probability that they had been obtained by chance (Table 1). The combination of the top 50 ranking residues associated with the selected centrality measures led to the identification of 92 residues, of which 30 (with a significance *p*-value of 1.1 × 10^−5^) were located in the binding pockets.

A visual comparison of the RIN and GRID analyses of the protein binding pockets is shown in Figure 3. It is worth noting that for pockets 3 and 4, the topological analysis failed to identify many sites, although a lot of the identified residues were in the neighboring areas. However, for pockets 1 and 2, a statistically high number of residues (32% and 44%, respectively) was identified, confirming the goodness of the method. Moreover, for those two pockets, the recognition percentages greatly increase if neighboring sites are considered.

### 2.4. In Silico Drug Repurposing

By applying a drug repurposing approach, we carried out a structure-based virtual screening of the FDA-approved drugs for human use and investigational molecules against the four druggable SARS-CoV-2 helicase binding pockets with the aim of identifying promising compounds with a multi-site profile. As illustrated in Figure 4, we selected 61, 51, 468, and 543 molecules related to pockets 1, 2, 3, and 4, respectively, according to their G-score ranking (best G-score > 2 kcal/mol from the minimum).

As reported in Table 2, we found 14 shared compounds that are able to recognize there out of four binding sites. We firstly applied known criteria during the visual inspection [27], such as improper atom types, shape complementarity, hydrophilic–hydrophobic mismatches, hydrogen bonding network, hydrophobic interactions, π-stacking, and distorted ligand geometry. All compounds provided an acceptable picture of the interaction network in each selected pocket (Appendix A). Moreover, due to the lack of experimental SARS-CoV-2 nsp13 co-crystallized models for pockets 2, 3, and 4, we decided to encompass all 14 screened molecules. At this stage, we considered their chemical diversity, subsequently classifying them into eight different groups (Table 2); for each one, we checked the commercial availability, the literature data, and the cost-effectiveness. Based on the above-mentioned reasons, we purchased the five compounds reported in Table 3 for further investigation in terms of their antiviral activity.

In analyzing the binding mode of the five compounds in pocket 2 (Figure 5 and Appendix A), it was observed that PF-00610355, an orally inhaled β_2_ adrenoreceptor agonist investigated for the once-daily treatment of Chronic Obstructive Pulmonary Disease (COPD) [31], showed the lowest G-score value. In detail, it was involved in several H-bond (HB) interactions with Asn179, Leu412, Gly415, Leu417, Ser485, and Asn557 residues and in a π–π interaction with Tyr515. Such a tyrosine residue is part of the RNA/DNA binding channel of the helicase and was found to be in direct contact with some fragments that are particularly attractive as starting points for the design of RNA competitive inhibitors [24].

NADH, a coenzyme useful as an electron carrier due to its oxidized (NAD^+^) and reduced (NADH) forms, was found to interact with Pro408, Thr410, Leu412, Thr413, Gly415, Asp534, and Arg560. Ceftaroline fosamil, an advanced-generation, parental cephalosporin, had four HB interactions with Lys146, Asn179, His311, and Ala312. PF-03715455, a p38 kinase inhibitor [32], was able to bind Leu417, Ser485, and Arg560, while polydatin, a natural precursor and glycoside form of resveratrol [30], engaged four HB interactions with Asn179, Leu417, and Asp534.

Concerning pocket 3, most of the five shared compounds (Figure 6 and Appendix A) were stabilized by an interaction network within the Arg-rich region located at position 15, 21, 22, and 129. In addition to these residues, Glu136 anchored polydatin, NADH, PF-03715455, and PF-00610355, while Phe24 formed a π–π interaction with both NADH and PF-03715455. Ala4 bound both polydatin and ceftaroline fosamil, with the latter also involved in an additional HB with Leu235.

Regarding pocket 4 (Figure 7 and Appendix A), all the investigated compounds recognized this binding cavity well. In particular, PF-00610355 formed six HB interactions with Gly54, Cys50, Asn51, Asn46, and Gly67, and a π–π interaction with Phe90. PF-03715455 established two HB with Ala52 and Asn46, and two halogen bonds with Cys55. NADH anchored to Ala52, Cys55, Asn46, Ser69, Tyr70, and Ser44, while polydatin formed three HB with Tyr48 and Gly67. Finally, ceftaroline bound Val45, Tyr70, and Lys94 by means of HB and salt-bridge interactions, respectively. Most of these interactions involve multiple residues that are universally conserved in the α- and β-coronaviruses [20].

### 2.5. In Vitro Evaluation of Compounds Enzymatic Activity

The most promising compounds were assessed in both helicase-associated enzymatic activities. Compounds SSYA10-001 and Licoflavone C, reported to inhibit SARS-CoV-2 nsp13 [33,34], were used as positive controls (Table 4). The results showed that the compound PF-03715455 was the most potent inhibitor capable of blocking both nsp13 enzymatic functions in the low micromolar range (IC_50_ values of 3.02 μM against the unwinding and 9.26 μM against the NTPase activity). PF-00610355 blocked the unwinding activity, with an IC_50_ of 22.4 μM, as it is not active against NTPase activity.

## 3. Discussion

The high degree of conservation characterizing the SARS-CoV-2 helicase, which emerged from our analysis of >5 million SARS-CoV-2 sequences, defines this viral protein as an ideal target for antiviral therapy that is meant to thwart the genetic evolution that SARS-CoV-2 is undergoing. Notably, we also demonstrated that a particularly high extent of conservation (>99%) characterizes helicase residues, which is crucial for their enzymatic activity and for the binding of the antiviral compounds identified in this study, thus emphasizing their potential to guarantee a wide-spectrum effectiveness against all SARS-CoV-2 variants. Previous evidence has also highlighted that helicase is the most conserved non-structural protein within the entire Coronaviridae family [35,36]. Therefore, compounds targeting helicase, which are characterized by a potential broad-spectrum coronavirus activity, could be critical in facing novel emerging coronaviruses.

Despite the de novo development programs applied to crucial SARS-CoV-2 targets have allowed to identify some potent inhibitors [37,38,39], the exorbitant costs, high attrition rate, and extensive periods of time for market approval tend to discourage research in this direction. Therefore, the drug repurposing of approved or investigational drugs is still an attractive proposition, especially in the case of an emerging, infectious viral disease. From the research perspective, such an approach is more advantageous as compared to de novo drug discovery. First, the development risk is significantly lower, and it might be possible to bypass preclinical trials. Moreover, once a suitable repurposing candidate has been identified, the pharmaceutical companies will be able to cost-effectively start large-scale production for emergency use. Finally, by combining the repurposed molecules, monotherapy resistance could be significantly reduced. At the same time, compounds with novel antiviral properties could be used as molecular tools to unravel the mechanism and pathogenesis of emerging viruses [40]. In this regard, remdesivir and molnupiravir, the small-molecule SARS-CoV-2 RdRp inhibitors that had received approval or emergency-use authorization to treat COVID-19 in several countries, were originally developed for other viral infections [9,41].

Our in silico drug repurposing approach allowed us to successfully identify PF-03715455 and PF-00610355, kindly provided by Pfizer Inc., as novel helicase inhibitors, with PF-03715455 capable of blocking both the unwinding and NTPase activities of nsp13. Both Pfizer investigational compounds have already been theoretically predicted as interesting compounds for the inhibition of different SARS-CoV-2 targets [42,43]. Our structural insights on PF-03715455 support the previous hypothesis, thus rationalizing the possible therapeutic option of using p38 inhibitors in serious COVID-19 infection under clinical trials [44]. More specifically, and to our knowledge, it is the first time that the enzymatic profile of PF-03715455 and PF-00610355 has been reported on a poorly explored target such as the conserved helicase. The tendency of the other SARS-CoV-2 non-structural proteins to easily mutate and the pivotal role of nsp13 in the replication machinery could focus the efforts of antiviral research towards this druggable target. Interestingly, the recent approval of nirmatrelvir (PF-07321332), co-administered with a low dose of ritonavir, may open the possibility of evaluating its potential synergistic effect with these novel nsp13 inhibitors.

The antiviral therapeutic potential of both Pfizer investigational compounds is still more interesting in view of the fact that they have been used in clinical trials for pulmonary diseases. Given the emergent need for treatment against SARS-CoV-2, all these considerations drive us to propose immediate trials in order to evaluate the repurposing of these two molecules as antiviral agents.

## 4. Materials and Methods

### 4.1. Conservation Analysis

The SARS-CoV-2 genome N = 5,156,608 sequences covering 1692 sublineages (including 120,556 sequences of 98 Omicron sublineages) were collected until 17 March 2022 from the GISAID database and used to accurately define any amino acid substitution in the helicase. By applying a stringent quality filter, only entire sequences with high quality were included in this study. High-quality sequences were defined as genomes of >29,000 nucleotides, characterized by the presence of <1% ambiguous nucleotides and <0.05% unique amino acid mutations, and with no insertions or deletions unless verified in the sequence by the submitter. The Bioedit software and MAFFT server were used to align sequences against the reference sequence (NC_045512.2 SARS-CoV-2-Wuhan-Hu-1 isolate). Then, the alignments were split into different proteins to set apart helicase sequences. Sequences with a mixture of wild-type and mutant residues at single positions were considered to have the mutant(s) at that position. Finally, based on alignment conservation annotation, we measured the similarities and differences from the individual amino acid level to the sequence level by measuring the number of conserved amino acids for each column (position) of the alignment. To standardize the effect of the sequence, the prevalence of genetic conservation was calculated based on total representative sequences while considering the sample size and margin of error; a 99% confidence interval was thus set as the limit.

### 4.2. Molecular Modeling

For the modeling studies, we used the crystal structure of the SARS-CoV-2 helicase in complex with the ATP analog AMP-PNP, deposited in the Protein Data Bank (PDB) [45] with the PDB code 7NN0 [24]. The receptor structure was prepared by using the Protein Preparation Wizard tool implemented in Maestro using OLPS-2005 as the force field [46,47]. Residual crystallographic buffer components and water molecules were removed, except for three water molecules that were coordinated with the Mg^2+^ atom; missing side chains were built using the Prime module. Hydrogen atoms were then added, side chain proto-nation states at pH 7.4 were assigned, and an energy minimization simulation was performed. The SARS-CoV-2 helicase binding sites were identified by using the GRID 2021 interface in the classical GRID software (Molecular Discovery Ltd., Borehamwood, United Kingdom) [23]. Each generated cavity was adopted as the starting point for further molecular recognition studies from the DrugBank database [48]. The library contained 3662 FDA compounds, including both approved drugs for human use and investigational molecules, and was prepared by means of the LigPrep Tool [49]; hydrogens were then added, salts were removed, and ionization states were calculated using Epik at pH 7.4. Each structure was submitted to energy minimization steps using OPLS_2005 as force field, as implemented in MacroModel v. 11.9 (Schrödinger, LLC, New York, NY, USA) [50]. The virtual screening was carried out by means of the Glide software v. 7.8 (Schrödinger, LLC, New York, NY, USA), using the SP (standard precision) algorithm [51] and running 10 poses per ligand for each identified site. To generate the grid box, we specified the residues of each cavity (Appendix A), and we pointed to the related centroid with a size of 30 × 30 × 30 Å. In order to filter the scored compounds, we selected them from within 2 kcal/mol of the minimum G-score value for each analyzed pocket. Finally, we examined only those compounds capable of recognizing at least 3 of the 4 binding sites. By means of visual inspection and based on their commercial availability, we selected and purchased the 5 best *hits* for the biological tests. The binding modes of all the best-selected compounds and their interacting pattern in the active site of the analyzed targets were highlighted using a ligand interaction diagram in Maestro GUI (Schrödinger, LLC: New York, NY, USA) [52], while all 3D figures were generated by using Pymol v. 2.0.7 (Schrödinger, LLC: New York, NY, USA) [53].

### 4.3. Structural Bioinformatics Analysis

Network analysis has been shown to be a powerful instrument for the study of biological systems. In particular, the analysis of Residue Interaction Networks (RINs) has proven to be a very useful tool for highlighting those residues with crucial roles in protein functionality [26]. This analysis dates back to the end of the last century [54], and over the years, it has carved out an increasingly important part in molecular biology computational research [26]. Starting from the crystal structure of the SARS-CoV-2 helicase in complex with the ATP analog AMP-PNP (PDB ID 7NN0), three different methods to generate the RIN of the helicase were used. The first, which is the more standard method, is based on the C-alpha distance of amino acids (below the threshold of 8 Å, the amino acids are considered to be linked), while the second is based on the contacts between the residue atoms of the helicase. For both methods, the Cytoscape 3.9.1 software was used and was linked by StructureViz2 1.0.6 to the UCSF Chimera 1.16 environment. The last method is based on probabilistic residue interaction [55], and the computation of the RIN is available online on the website ring.biocomputing.it/submit. Different features can be considered when dealing with RIN, topological information (as centrality measures), geometrical properties (as clustering analysis), or dynamical process (Markov process, Network diffusion). Here, the analysis was limited to centrality measures for their efficacy in determining those residues with a key role in the function of a protein. Many different centrality measures were computed for the three considered RINs. It was clearly seen that the RIN obtained by means of the contacts between the residue atoms was able to characterize the binding pockets much better than the other two RINs. In particular, four centrality measures, due to their characteristics, proved to be the most relevant for the identification of residues belonging to the binding regions of enzymes: RCA (Radiality Centrality Analysis), BCA (Betweenness Centrality Analysis), CCA (Closeness Centrality Analysis), and Z-RCA (zeta-score of Residue Centrality Analysis) [56,57]. Finally, a statistical enrichment analysis based on the hypergeometric distribution, was used in order to perform the statistical significance test for the obtained results [58].

### 4.4. SARS-CoV-2 nsp13 Expression and Purification

The SARS-CoV-2 nsp13 his-tagged was expressed in E. coli BL21 Rosetta 2 cells from a pNIC-ZB (addgene #159614). A purification protocol was adopted from Newman et al. [24]. Terrific Broth media was used to culture the cells at 37 °C and at 200 rpm until 1.6 OD600. Then, protein expression was induced with 300 mM IPTG for 16 h at 18 °C, at 200 rpm, and then pelleted and resuspended in lysis buffer (50 mM HEPES pH 7.5, 500 mM NaCl, 5% Glycerol, 10 mM Imidazole, 0.5 mM TCEP, Merck Protease inhibitor cocktail III, 1:500). After sonication and clarification at 15,000× *g* for 30 min, the crude extract was used for a batch binding of 40 min in a three-dimensional rotating mixer with 5 mL of Ni-sepharose (cytiva) under orbital. Then, the tubes were centrifuged at 700× *g* at 4 °C for 5 min, and the resin was loaded onto a gravity flow column (econo column BioRad) connected to a BioLogic LP (BioRad). It was subsequently washed (1 mL/min) with a lysis buffer for 40 min and with a wash buffer (50 mM HEPES pH 7.5, 500 mM NaCl, 5% Glycerol, 45 mM Imidazole, 0.5 mM TCEP) for 25 min. A further 10 min wash with a Hi-salt buffer (50 mM HEPES pH 7.5, 1 M NaCl, 5% Glycerol, 0.5 mM TCEP) was performed, followed by another 10 min of wash buffer. Proteins were eluted with the addition of 15 mL of elution buffer (50 mM HEPES pH 7.5, 500 mM NaCl, 5% Glycerol, 300 mM Imidazole, 0.5 mM TCEP). The elution fraction was analyzed for purity by SDS page, and clean fractions were pooled and dialyzed against a desalting buffer containing 25 mM HEPES pH 7.5, 250 mM NaCl, 20% Glycerol, and 0.25 mM TCEP. Proteins were stored at −80 °C.

### 4.5. Determination of SARS-CoV-2 nsp13 Unwinding-Associated Activity

The SARS-CoV-2 nsp13 unwinding-associated activity was measured in black 384-well plates (PerkinElmer). The reaction mixture, as described in [34], contained 20 mM Tris–HCl pH 7.2. 50 mM NaCl, 2 mM MgCl_2_, 10 μg/mL of BSA, 180 μM TCEP, 2 μM Hel Capture oligo (5′- TGG TGC TCG AAC AGT GAC -3′) (Biomers) 5% DMSO or inhibitor, and 1 nM of purified nsp13. The enzyme was pre-incubated for 10 min with the inhibitor at room temperature, and then reaction was started by adding 1 mM ATP and 750 nM annealed DNA substrate (5′- AGT CTT CTC CTG GTG CTC GAA CAG TGA C-Cy3-3′. 5′- BHQ-2-GTC ACT GTT CGA GCA CCA CCT CTT CTG A-3′) (Biomers). Reaction volume was 40 μL. Compounds SSYA10-001 and Licoflavone C were used as positive controls of inhibition; the reaction lasted for 15 min at 37 °C, and then the products were measured with Victor Nivo (Perkin) at 530/580 nm.

### 4.6. Determination of SARS-CoV-2 nsp13 ATPase-Associated Activity

The SARS-CoV-2 nsp13 helicase-associated activity was measured in a transparent 96-well plate (PerkinElmer), as described in [34]. Briefly, the reaction mixture contained 20 mM Tris–HCl pH 7.2, 50 mM NaCl, 2 mM MgCl2, 10 µg/mL of BSA, 180 µM TCEP, 5% DMSO or inhibitor, and 25 nM of purified nsp13. The reaction was started by adding 400 µM ATP. Total reaction volume was 25 μL. Compound Licoflavone C was used as positive control of inhibition. The reaction lasted for 30 min at 37 °C. Then, 50 µL of a stop/revealing solution was added (Biomol^®^ Green Reagent, Prod. No. BML-AK111. Enzo Lifescience), and the reaction was incubated for 10 min at RT and protected from light. The products were measured with Victor Nivo (Perkin) at 650 nm.

## 5. Conclusions

In this work, we assessed the conservation of nsp13 and the druggability of its binding pockets. Our analysis highlighted the high degree of conservation of the helicase, particularly in its domains crucial for enzymatic activity. This study represents a successful example of in silico drug repurposing, by which we identified structurally novel nsp13 inhibitors from among FDA-approved and investigational compounds. Two valuable compounds, PF-03715455 and PF-00610355, were found to inhibit the enzyme with IC_50_ values in the micromolar range. Most of the interacting residues of pockets 2 and 4 were highly conserved throughout human coronaviruses [20,59], thus suggesting that the virus is unlikely to develop resistance to both compounds. Our multi-site approach evidences the capability of PF-03715455 to block both unwinding and NTPase activities, thus providing an additional scaffold for further medicinal chemistry optimization.

## Figures and Tables

**Figure 1 molecules-27-07522-f001:**
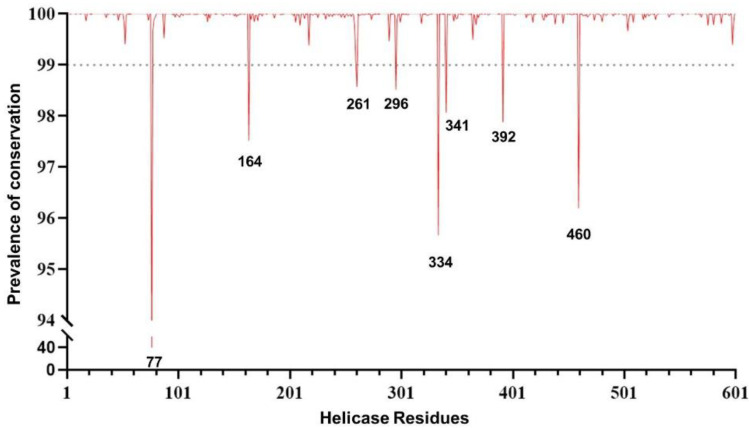
Genomic plot showing the prevalence of the genetic conservation of the SARS-CoV-2 helicase (nsp13).

**Figure 2 molecules-27-07522-f002:**
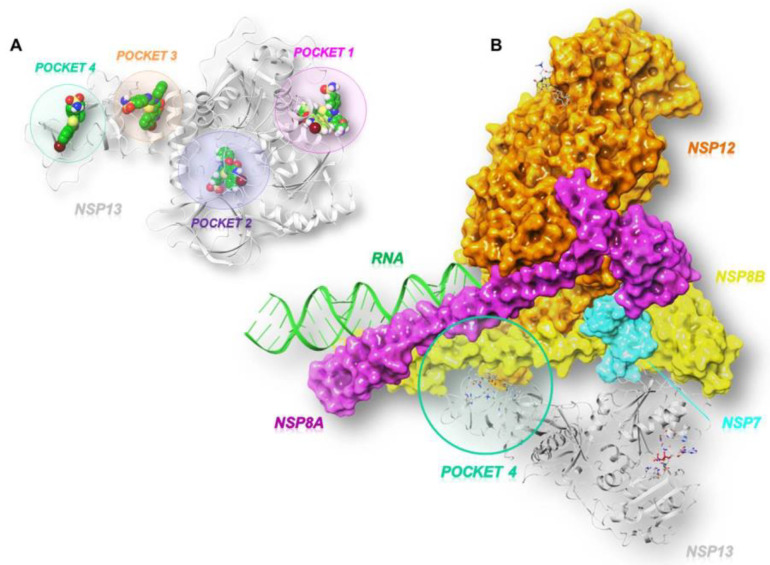
Three-dimensional representation of (**A**) the three SARS-CoV-2 helicase (PDB code:7NN0) [21] binding pockets (1 in pink, 2 in violet, and 3 in orange) obtained by GRID analysis, and (**B**) a focus on the interface site (pocket 4 in green) between SARS-CoV-2 nsp13 and nsp7-nsp8-nsp12 (PDB code:6XEZ) [20].

**Figure 3 molecules-27-07522-f003:**
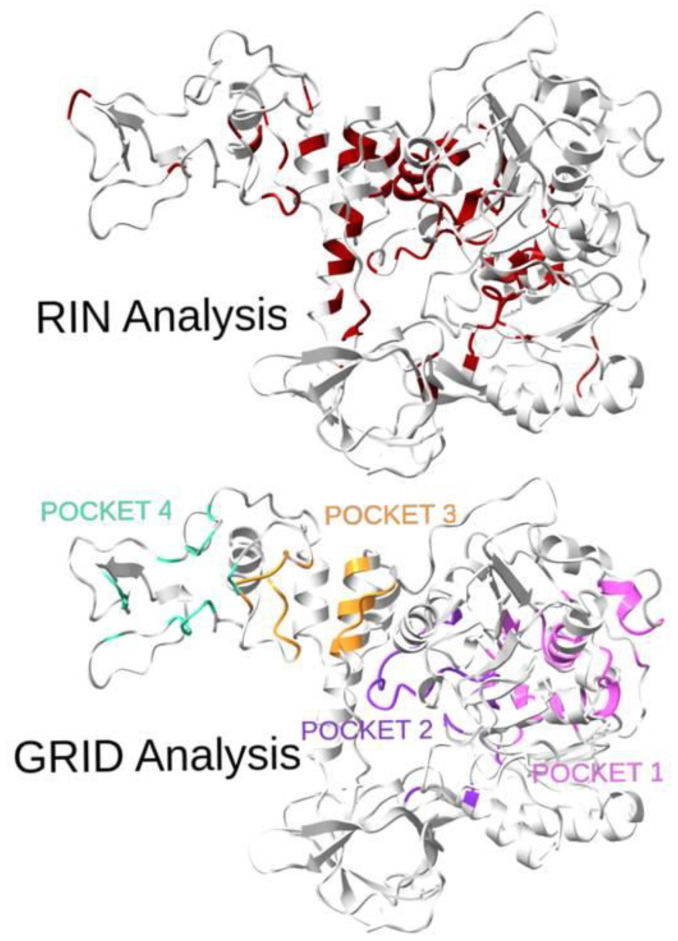
Three-dimensional representation of the SARS-CoV-2 helicase with highlighted residues from the topological analysis (top: RIN Analysis, best residues determined by centrality measures are in red) and residues from the four pockets (bottom: GRID Analysis, 1 in pink, 2 in violet, 3 in orange, 4 in green). The overlap of the two sets of residuals is quite evident and is confirmed by the quantitative results reported in Table 1.

**Figure 4 molecules-27-07522-f004:**
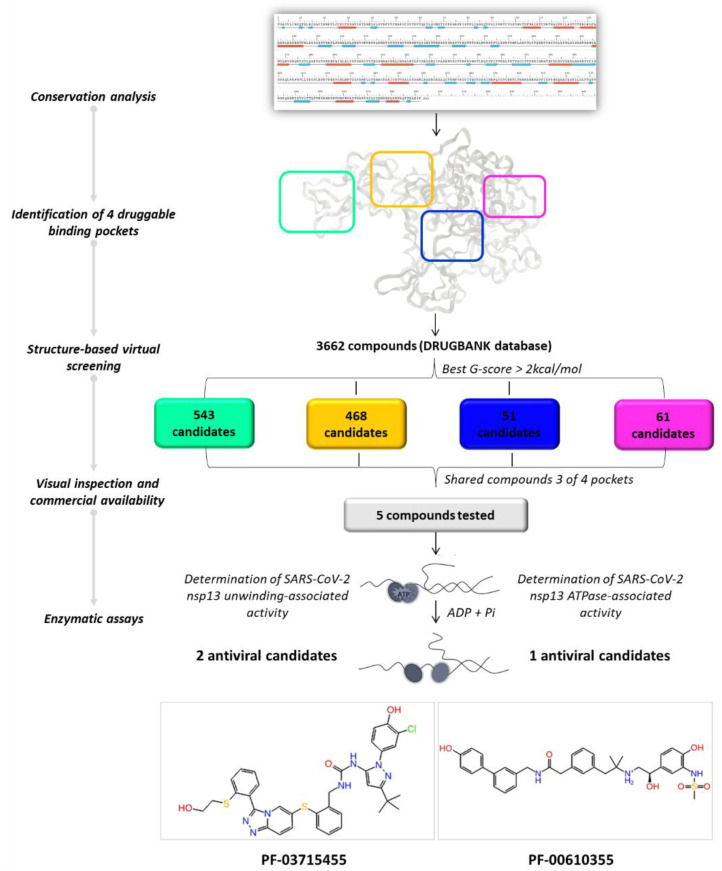
Graphical workflow of the applied structure-based virtual screening approach.

**Figure 5 molecules-27-07522-f005:**
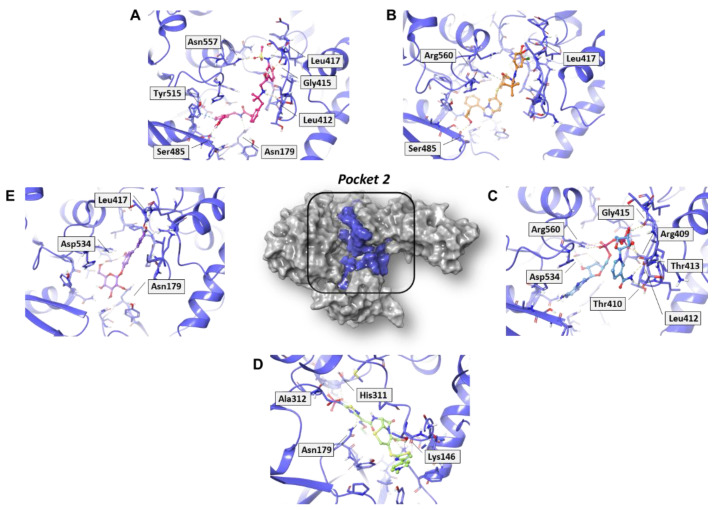
Three-dimensional representation of the best docking pose of (**A**) PF-00610355 (magenta carbon sticks), (**B**) PF-03715455 (orange carbon sticks), (**C**) NADH (cyan carbon sticks), (**D**) ceftaroline fosamil (green carbon sticks), and (**E**) polydatin (violet carbon sticks) in pocket 2 of the SARS-CoV-2 nsp13. The enzyme and the residues that have crucial contacts with the compounds are shown as violet cartoon and violet carbon sticks, respectively.

**Figure 6 molecules-27-07522-f006:**
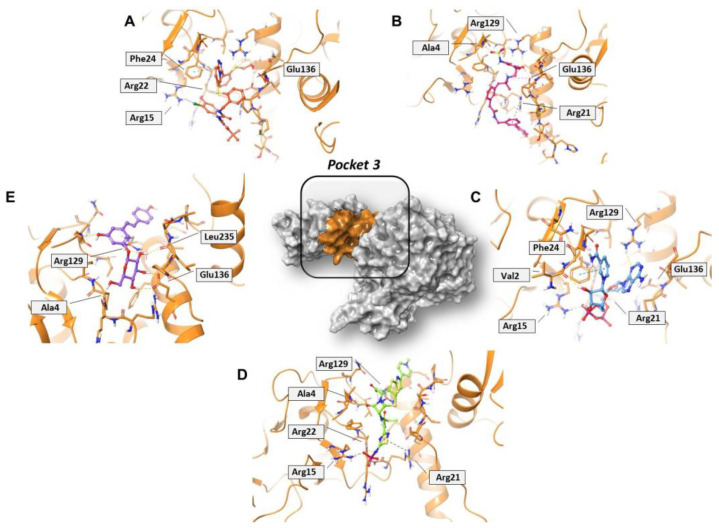
Three-dimensional representation of the best docking pose of (**A**) PF-00610355 (magenta carbon sticks), (**B**) PF-03715455 (orange carbon sticks), (**C**) NADH (cyan carbon sticks), (**D**) ceftaroline fosamil (green carbon sticks), and (**E**) polydatin (violet carbon sticks) in pocket 3 of the SARS-CoV-2 nsp13. The enzyme and the residues that have crucial contacts with the compounds are shown as orange cartoon and orange carbon sticks, respectively.

**Figure 7 molecules-27-07522-f007:**
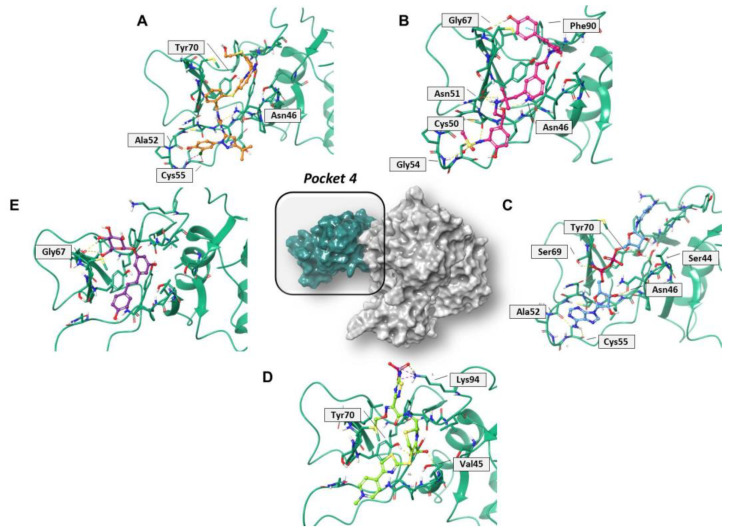
Three-dimensional representation of the best docking pose of (**A**) PF-00610355 (magenta carbon sticks), (**B**) PF-03715455 (orange carbon sticks), (**C**) NADH (cyan carbon sticks), (**D**) ceftaroline fosamil (green carbon sticks), and (**E**) polydatin (violet carbon sticks) in pocket 4 of the SARS-CoV-2 nsp13. The enzyme and the residues that have crucial contacts with the compounds are shown as green cartoon and green carbon sticks, respectively.

**Table 1 molecules-27-07522-t001:** Number of residues in the intersection between different numbers of top-ranking residuals (first column) associated with various centrality measures (first row) such as Residue Centrality Analysis (RCA), Betweenness Centrality Analysis (BCA), Closeness Centrality Analysis (CCA), and those belonging to the four pockets. In the parentheses, the p-value associated with the relevance of the intersection value.

	RCA	BCA	CCA	Z-RCA
**10**	5 (2.1·10^−4^)	6 (2.5·10^−4^)	5 (2.1·10^−4^)	6 (2.5·10^−4^)
**20**	10 (8.3·10^−5^)	9 (5.1·10^−4^)	10 (8.3·10^−5^)	9 (5.1·10^−4^)
**50**	19 (3.7·10^−5^)	20 (8.9·10^−6^)	19 (3.7·10^−5^)	18 (1.4·10^−4^)

**Table 2 molecules-27-07522-t002:** List of the 14 compounds identified by means of the in silico drug repurposing approach against SARS-CoV-2 helicase capable of recognizing three out of four enzyme-binding pockets. For each compound, the chemical group, the name, the DrugBank ID code, the 2D structure, and the Glide docking score (G-Score) values, which resulted from the molecular recognition of the four druggable sites, are reported. The compounds are in alphabetical order. The G-score values within 2 kcal/mol from the minimum for each pocket are highlighted in green.

Chemical Group	Name	DrugBank ID	Structure	Pockets
**1**(−11.73 ^a^)	**2**(−8.66 ^a^)	**3**(−6.82 ^a^)	**4**(−6.81 ^a^)
**Acetamide derivatives**	**PF-00610355**	**DB11871**	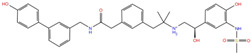	−6.19	**−8.51**	**−5.65**	**−6.81**
**Beta-lactam derivatives**	Cefpiramide	DB00430	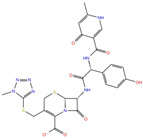	−5.48	−7.05	−5.51	−5.47
Ceftaroline fosamil	DB06590	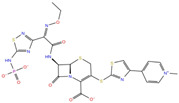	−9.43	**−6.82**	**−5.38**	**−5.15**
**Glicoside derivatives**	Acteoside	DB12996	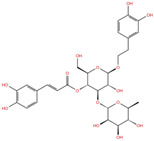	−6.05	−8.13	−4.83	−5.49
Polydatin	DB11263	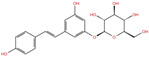	−5.94	**−6.70**	**−5.02**	**−4.89**
Rutin	DB01698	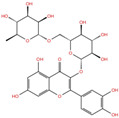	−5.62	−6.84	−5.61	−5.01
**Phenol derivatives**	Metaraminol	DB00610	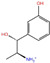	−4.92	−6.84	−6.18	−5.25
**Phosphono derivatives**	Foscarnet	DB00171	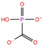	−11.73	−4.10	−4.85	−4.99
**Pteridine analogs**	5-methyltetrahydrofolic acid	DB04789	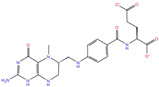	−10.21	−5.85	−4.99	−5.48
Riboflavin	DB00140	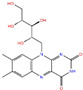	−5.80	−6.68	−4.86	−5.44
**Purine analogs**	Inarigivir soproxil	DB15063	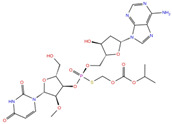	−7.13	−7.79	−6.49	−5.49
NADH	DB00157	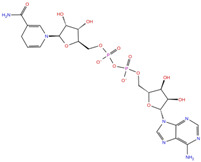	−8.60	**−7.15**	**−5.41**	**−5.38**
Regadenoson	DB06213	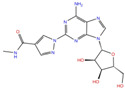	−5.29	−6.92	−5.15	−5.29
**Triazole derivatives**	PF-03715455	DB12138	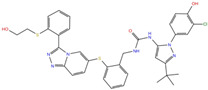	−6.42	**−6.82**	**−4.86**	**−5.86**

^a^ This value indicates the absolute best G-score value for each analyzed pocket and is expressed in kcal/mol.

**Table 3 molecules-27-07522-t003:** Name, developmental phase, and pharmacological data of the five purchased compounds selected for antiviral activity experiments on the SARS-CoV-2 helicase.

Name	Developmental Phase	Pharmacological Data
Ceftaroline fosamil	Approved	Antibacterial activity [28]
NADH	Approved Nutraceutical	Nutritional and vitamin supplementation [29]
PF-03715455	Investigational	In clinical trials for the treatment of asthma, pulmonary disease, chronic obstructive, and Chronic Obstructive Pulmonary Disease (COPD) ClinicalTrials.gov (NCT02219048)
PF-00610355	Investigational	In clinical trials for the treatment of lung disease, pulmonary disease, asthma, and bronchial diseases ClinicalTrials.gov (NCT00783406)
Polydatin	Approved	Anti-inflammatory, immunoregulatory, anti-oxidative, and anti-tumor activities [30]

**Table 4 molecules-27-07522-t004:** Inhibition of SARS-CoV-2 nsp13 helicase-associated activities by compounds.

Compound	^a^ IC_50_ (μM) Unwinding	^b^ IC_50_ (μM) NTPase
Ceftaroline fosamil	>30 (100%) ^c^	>30 (100%)
NADH	>30 (100%)	>30 (100%)
Polydatin	>30 (100%)	>30 (100%)
PF-03715455	3.02 ± 0.21	9.26 ± 1.93
PF-00610355	22.4 ± 1.7	>30 (100%)
SSYA10-001	2.3 ± 0.7	ND ^d^
Licoflavone C	8.7 ± 1.3	20.9 ± 0.8

^a^ Compound concentration required to inhibit the SARS-CoV-2 nsp13-associated unwinding activity by 50%; ^b^ Compound concentration required to inhibit the SARS-CoV-2 nsp13-associated ATPase activity by 50%; ^c^ Percentage of control measured in the presence of the highest tested compound concentration; ^d^ Not tested.

## Data Availability

Not applicable.

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
