# Peer review of "Targeting SARS-CoV-2 nsp13 Helicase and Assessment of Druggability Pockets: Identification of Two Potent Inhibitors by a Multi-Site In Silico Drug Repurposing Approach"

_molecules, 2022, doi:10.3390/molecules27217522_

Round 1

Reviewer 1 Report

Comments on the manuscript ID molecules-1951720 by Romeo et al. titled “Targeting SARS-CoV-2 nsp13 helicase and assessment of druggability pockets: identification of two potent inhibitors by a multi-site in silico drug repurposing approach”:

The authors provide a brief review of various drugs with different mechanisms of action being investigated for potential treatment of COVID-19. Taking in mind that the drug repurposing of approved or investigational drugs represent a practical approach for the fast identification, characterization, and deployment of antiviral treatments thanks to the availability of existing detailed data on drug medicinal chemistry, human pharmacology, and toxicology, they perform a conservation analysis of a SARS-CoV-2 helicase (nsp13 helicase) and apply computational and bioinformatics tools to predict the selected enzyme binding pockets interactions with known drugs. Starting from a database of around 4000 drugs already approved, they select 14 compounds able to recognize 3 out of 4 sites/pockets. The problem is that these 14 compounds differ significantly in chemical structure and the selection (‘by visual inspection analysis’) of the 5 promising compounds seems unmotivated. What criteria are used to select suitable candidates?

Reviewer 2 Report

1.      The SARS-CoV-2 non-structural protein 13 (nsp13) helicase is an essential enzyme for viral replication and has been identified as an at-tractive target for developing of new antiviral drugs. Correct this sentence.

2.      the drug repurposing of approved or investigational drugs could represent a  practical approach for the fast identification, characterization, and deployment of antiviral treatments thanks to the availability of existing detailed data on drug medicinal chemistry, human pharmacology, and toxicology. Adding this literature shall be helpful https://doi.org/10.1042/BSR20201256

3.      Introduction section can be improved by adding useful information from this publication.

https://doi.org/10.1016/j.ijbiomac.2021.02.071

4.      In silico drug repurposing. This section needs a rewriting in a crisper manner exactly explaining how the study was proceeded. Authors can summarise their virtual screening and selection of compounds via graphical image that will aid in easy understanding of the readers. The authors quote “According to their G-score ranking (best 185 G-score > 2 kcal/mol), we selected 61, 51, 468 and 543 molecules related to pockets 1, 2, 3 186 and 4, respectively.” Finally, by visual inspection and based on their commercial availability, we purchased the 5  compounds reported in Table 3 to be further investigated in terms of their antiviral activity. What does authors meant by visual inspection?

5.      Why 2D analysis was not shown?

6.      Overall the study is significant in the field of COVID19 therapeutics but need few improvements and after these improvements the manuscript can be accepted for publication.

7.      The English Language need to be improved restructuring long sentences and removing all the grammatical flaws.

.

Round 2

Reviewer 1 Report

The criteria listed by the authors in their reply (chemical diversity, novelty, and commercial availability of the best scored ligands) do not correspond to the criteria indicated by Fischer et al. (André Fischer, Martin SmiesÌŒko, Manuel Sellner, and Markus A. Lill, “Decision Making in Structure-Based Drug Discovery: Visual Inspection of Docking Results”, J. Med. Chem. 2021, 64, 24892500) as successful in the visual inspection of docking results. Meaningful criteria should be used to select compounds. These criteria should be clearly indicated in the article.

Author Response

We are sorry for this misunderstanding, we did not mean that the criteria listed in our reply corresponded to those reported by Fischer et al. We wanted to highlight the importance of the visual inspection after docking simulations.

According to the reviewer suggestion, we implemented the manuscript as follows: “We firstly applied known criteria during visual inspection [26], such as improper atom types, shape complementarity, hydrophilic-hydrophobic mismatches, hydrogen bonding network, hydrophobic interactions, π-stacking and distorted ligand geometry. All compounds provided an acceptable picture of the interaction network in each selected pocket (Figures S2-S4). Moreover, due to the lack of experimental SARS-CoV-2 nsp13 co-crystallized models for pockets 2, 3 and 4, we decided to encompass all 14 screened molecules. At this stage, we considered their chemical diversity, thus classifying them into 8 different groups (Table 2) and, for each one, we checked the commercial availability, the literature data and the cost-effectiveness. Based on the above mentioned reasons, we purchased the 5 compounds reported in Table 3 to be further investigated in terms of their antiviral activity.”

Moreover, we modified Table 2 by including the classification of the 14 screened compounds based on their chemical scaffold and we added the best docking poses of all compounds versus the 3 nsp13 pockets in the Supplementary Material (Figures S2-S4).